Genetic diversity analysis of the natural regeneration loci of Liriodendron chinense in artificial mixed forests in the rocky desertification area of Western Hunan

Jia Ziying 1
Wang Yixuan 1
Huang Bingkun 2
Liang Miao 3
Ge Chong 1
Zhu Ninghua 1
You Ren ren.you@wsu.edu 4 5
1 Central South University of Forestry and Technology , Changsha , Hunan province , China
2 Chenshan Forest Farm, Anfu County , Jian , Jiangxi , China
3 Xiangxi Forest Ecology Research Experiment Station, Yongshun County , Jishou , Hunan , China
4 College of Life and Environmental Science, Central South University of Forestry and Technology , Changsha , Hunan , China
5 School of the Environment, Washington State University , Pullman , Washington state , United States of America
Brygadyrenko Viktor
Electronic publication date: 2025 Oct 23
Publication date: 2025
Volume: 13
Electronic Location ID: e20138
Received 2025 Jan 27; Accepted 2025 Sep 4
Copyright: ©2025 Jia et al.
Copyright year: 2025
Copyright holder: Jia et al.
License: This is an open access article distributed under the terms of the Creative Commons Attribution License, which permits unrestricted use, distribution, reproduction and adaptation in any medium and for any purpose provided that it is properly attributed. For attribution, the original author(s), title, publication source (PeerJ) and either DOI or URL of the article must be cited.
License URL: https://creativecommons.org/licenses/by/4.0/

Keywords: Liriodendron chinense, Rocky desertification areas, SSR, Genetic diversity

Funding: Hunan Province Forestry Science and Technology Research and Innovation XLKY2023-30 This research was funded by Hunan Province Forestry Science and Technology Research and Innovation, grant number XLKY2023-30. The funders had no role in study design, data collection and analysis, decision to publish, or preparation of the manuscript.

==============================
Liriodendron chinense plays a crucial role in improving the ecological environment and combating soil erosion in the rocky desertification area of Western Hunan, China. However, there is still a lack of systematic research on the genetic diversity of natural populations of the L. chinense in rocky desertification areas. This study employed 11 simple sequence repeat (SSR) markers to analyze genetic diversity and spatial genetic structure in a population of 318 L. chinense individuals. We conducted parentage analysis on individuals from a limited area of natural regeneration to quantify pollen and seed-mediated gene flow separately. Based on diameter classification, L. chinense individuals in the large diameter class can be considered as potential parents. The results show that there is moderate genetic diversity in the natural populations of the L. chinense. The spatial genetic patterns of the adult individuals indicate that significant gene flow occurs primarily at short to medium distances, with about 70% occurring within a range of less than 80 m. Among the 318 L. chinense individuals analyzed, 201 were predominantly assigned to the parental generation, with 41 showing closest genetic similarity to the maternal parent. These results indicate that the majority of pollen (63.2%) originated from within the sampling area, which suggests a substantial proportion of natural regeneration occurred within the 2.5 hm2 stand. These findings further elucidate the natural regeneration process of L. chinense and provide a theoretical foundation for ecological restoration efforts in rocky desertification areas.

Introduction

The rocky desertification area in Western Hunan is known for its unique geographical and ecological environment (Luo et al., 2024). This region has been severely affected by land degradation and rocky desertification resulting from prolonged natural processes and anthropogenic activities (Chang et al., 2024). Western Hunan is the primary region involved in the Grain to Green Program (GTGP), with Liriodendron chinense being the principal species used for restoration (Xiao et al., 2023).

L. chinense is an ancient relict tree species of the tertiary period, which is also a precious second-level protected plant in China (Chen et al., 2024). Currently, only two species remain: L. chinense and Liriodendron tulipifera, which serve as a typical example of disjunct distribution between East Asia and North America (Xiang, Soltis & Soltis, 1998; Xiang et al., 2000). This holds significant research value for paleobotanical systematics (Maxwell et al., 2024). Owing to its wood properties, which include straight grain, fine texture, light weight, and softness, L. chinense is extensively utilized in the construction and furniture industries (Bai, Wang & Hu, 2024). However, the natural genetic diversity of L. chinense has significantly declined due to human-induced habitat degradation as well as inherent reproductive biological constraints. Consequently, natural regeneration has become limited, necessitating the development of alternative propagation methods and enhanced conservation efforts (Zhang & Ma, 2008).

High genetic diversity within a population enhances a species’ ability to withstand environmental and anthropogenic stressors, thereby increasing its resilience and adaptive potential (Milesi et al., 2024). The conservation of L. chinense is therefore critical for ensuring the species’ long-term viability amid ongoing environmental change. Achieving this goal requires a comprehensive understanding of its genetic diversity and population genetic structure (Casier et al., 2024; Xu et al., 2024). Furthermore, high genetic diversity confers a strong capacity for adaptation and evolutionary change (Lei et al., 2024).

DNA-based marker is an effective method for genetic characterization of plant and animal loci (Gupta, Rustgi & Mir, 2008). Simple sequence repeat (SSR) marker technology is widely used in genetic analyses owing to its advantages, which include the detection of multiple alleles, codominant inheritance, and high reproducibility. This technology enables the detection of abundant polymorphic loci across species and is extensively applied in plant genetic mapping and germplasm identification (Schie, Chaudhary & Debener, 2014). However, their application in wild tree species has been limited, primarily due to the lack of available markers. Owing to sequence homology in SSR flanking regions, these markers can be evaluated for cross-transferability and utilized in closely related species (Aiello et al., 2020).

For now, research on L. chinense has primarily focused on its phylogenetic positioning, morphology, distribution range, habitat range, physiological ecological traits, and the genetic diversity of seed orchards. However, studies on the genetic diversity and parentage relationships of large-scale natural regeneration cohorts of L. chinense in rocky desertification areas have yet to be reported. With above background, we used SSR marker technique to understand the genetic diversity, parent identification, and gene flow of the natural regeneration loci in L. chinense loci of the Western Hunan.

The study was aimed (1) using SSR marker technology to assess the genetic diversity of the regeneration loci of L. chinense; (2) to reveal the spatial dynamic change of genetic diversity of L. chinense, (3) assess the effects of habitat heterogeneity in karst rocky desertification regions on the spatial genetic structure, genetic diversity, genetic differentiation, genetic structure, and gene flow of L. chinense. This study represents the first application of SSR marker technology to investigate the genetic diversity of secondary L. chinense population in rocky desertification areas. The findings provide valuable insights for forest resource management in karst regions.

Materials & Methods

Plant materials and DNA extraction

The site located in the Wuling Mountain area of southwest China, serving as the national long-term research base for comprehensive management of rocky desertification (29°3′4″N, 110°13′35″E, 467 m a.s.l.). We measured the diameter at breast height (DBH) of L. chinense with a diameter ≥ one cm in the plot and recorded their coordinates by using real-time kinematic technology (RTK). And we utilized ArcGIS (Esri, Redlands, CA, USA) software to create a population L. chinense distribution map based on individual distribution coordinates (Fig. 1).

We collected 318 one-year-old leaves from tagged L. chinense individuals, including a few L. chinense trees artificially planted in 1978 and their naturally regenerated individuals. Leaf tissues were stored at −80 °C for subsequent DNA extraction and were ground into powder using liquid nitrogen. Genomic DNA was extracted from L. chinense using the DP320 DNA Secure Plant Genomic DNA Extraction Kit (TIANGEN DNAsecure Plant Kit DP320-03; TIANGEN, Beijing, China). The quality of the DNA samples was determined using a NanoDrop 2000 spectrophotometer (Thermo Fisher Scientific, Waltham, MA, USA) and agarose gel electrophoresis.

EST-SSR Primers, PCR Amplification, and microsatellite genotyping

The primers used in the experiment refer to the NCBI database expressed sequence tag-simple sequence repeat (EST-SSR) (http://www.ncbi.nlm.nih.gov/dbEST/index.html) primers and combined with the sequence information of SSR primers developed in published paper (n = 202) (Hirata et al., 2006). The target DNA fragments were amplified using the simple sequence repeat markers with tailed primer M13 (TP-M13-SSR) polymerase chain reaction (PCR) method (Supplement information).

The PCR amplification was performed using a thermal cycler (Bio-Rad, Hercules, CA, USA), and the PCR products were analyzed using an ABI 3730XL DNA analyzer (Applied Biosystems, Waltham, MA, USA). Sequencing was conducted using an ABI 3730 XL automated sequencer (Applied Biosystems) with LIZ-500 employed as the internal standard. Consequently, eleven pairs of highly polymorphic primers were selected for subsequent experimental research (Table 1).

Figure 1 Individual distribution of all Liriodendron chinense in sample plot.

Table 1 Information of SSR primers.

Locus	Primer sequence	Repeat motif	Annealing temperature (°C)	Allele range size (bp)	
LT002	CCTACCACCAGCAATACCTA	(GCA)9	56.6	190–199	
TCTCGTCGCTGAAGATATG	
LT031	TGAAGAACCCAACAACTCTC	(GA)18	54.2	211–217	
GTCGTAGCAGGTAGGTATGC	
LT058	TTAAAATGGAGGAACGAGAG	(GA)9	55.8	220–228	
GTAGAGGCTTCGAGTTTGTG	
LT089	GTCAGAGTGTTGGTTCAGGT	(TGA)7	55.3	263–274	
GGCAAAATGGGATTGTAAAG	
LT101	CCACAGGTTTTCTTCATTTC	(CT)10	53.5	343–353	
CGCATTGGATCTTCATCTTA	
LT113	CCAAGTGAAAATCAACTCCT	(CT)18	54.5	249–255	
ATCTCGACGGTGTTCTGAT	
LT119	CGGGAGGAATAGGACTAAAG	(AG)9	59.2	307–317	
GTGATGCTGCGATTTTCT	
LT125	GTCCAAGATCAAGGGTAGTG	(TC)15	56.8	280–300	
TAGATGGATTGACCCACTTG	
LT131	GCAGCATCTCCTCATATTCT	(AC)22	56.8	217–237	
TTGCAGTTGAGCTATTGTTG	
LT21	GGGCTTTAACCGAGGGATAG	(CT)16	58.8	241–265	
CATTTCCTGCCTCACATAGC	
LC36	GGAGGAAGCAAAAGGTCCGT	(TC)6	53.7	198–228	
AGCAAGGAGGCAGAGAGAGA	

Genetic diversity analysis

Peak profiles from capillary electrophoresis were read, verified, and genotyped using GeneMarker software version 2.6, generating a data matrix that served as the basis for subsequent analyses (Holland & Parson, 2011). We used GenALEx 6.5 software to analyze genetic diversity, the following data were obtained: number of alleles (NA), effective alleles (NE), Shannon index (I), observed heterozygosity (HO), expected heterozygosity (HE), genetic distance, and geographical genetic distance (D). And we used GenALex 6.5 software to calculate the fixation index (F), in-breeding coefficient within the population, genetic differentiation coefficient (FST), total inbreeding coefficient (FIT) for the final identified loci in the L.chinense population, while calculating gene flow (Nm) between loci using the formula Nm = (1−FST)/4 FST, and testing whether the selected 11 pairs of SSR primers deviate from Hardy-Weinberg equilibrium (HWE) (Khan et al., 2021). Polymorphic information content (PIC) was calculated using CERVUS 3.07 software (Kalinowski, Taper & Marshall, 2010). Invalid alleles (F(null)), also known as zero loci, were detected using Micro-Checker software (Van Oosterhout et al., 2004).

Analysis of spatial genetic structure

The spatial genetic structure (SGS) of L. chinense across different diameter classes and for the entire population within the plot was analyzed using SPAGeDi software version 1.5 (Sofletea et al., 2020). Pairwise kinship coefficients (Fij) for L. chinense individuals within the sample plot were calculated based on multiple loci, where Fij represents the probability of identity between the genotypes of two random individuals, i and j, sampled from the population (Loiselle et al., 1995). Based on the number and spatial distribution of individuals in the population, the distance classes were defined such that the number of individual pairs was approximately equal across all classes. Finally, 15 distance classes were divided into 10, 16, 22, 28, 34, 40, 46, 52, 58, 64, 70, 77, 85, 96 and 180 m. We calculated the average Fij value for each of the 15 distance classes, plot the regression of ln(rij) to obtain the regression slope bF (where rij represents the spatial geographical distance between samples i and j), confirm the linear positive correlation between Fij and ln(rij), and verify the significant deviation of genotype from random distribution through 10,000 permutations (Rousset, 2000). The strength of the SGS in L. chinense was assessed utilizing the Sp statistic, which represents the standard error of the regression of pairwise kinship coefficients on distance. The calculation formula for the Sp statistic is Sp = −bF/(1−F1), where bF is the linear regression slope of Fij on the natural logarithm of distance classes, and F1 is the average kinship coefficient of all pairs within the first distance class (Rousset, 1997; Vekemans & Hardy, 2004). In addition, for the analysis, each distance class must include data from more than 30 individuals to ensure the statistical validity of the analysis when considering diameter classes (Doligez & Joly, 1997). The L. chinense population was divided into five diameter classes (Table 2). Since the number of individuals in Class V is less than 30, Classes IV and V were combined for analysis.

Parentage analysis

The probability of identity (PID) and the probability of identity among siblings were calculated using GenALEx software version 6.5 (Peakall & Smouse, 2006). Geographical distances between individuals were calculated based on their spatial coordinates. These data were then used to statistically estimate dispersal distances for both seed and pollen flow. Parentage analysis was performed using CERVUS software version 3.07, which employs a maximum likelihood approach. It determines the parent that best matches the offspring by estimating the logarithm of the likelihood ratio (LOD) between the offspring and the parents. The parent with the highest LOD value is determined to be the most likely parent, while the simulator assesses the significance of the most likely parent based on the calculated critical value (Marshall et al., 1998). In parentage analysis with unknown parental sexes, the optimal parental combination is identified as the one with the highest and most significant offspring-parent trio LOD score (Trio LOD score). The relationship between gene flow (dg) and pollen dispersal (dp), ds is calculated as: dg2 = ds2+0.5dp2. Distances and probabilities of seed and pollen dispersal within a 180-meter radius were compared. Dispersal pattern maps for pollen and seed flow were generated based on the spatial coordinates of offspring, male parents, and female parents.

Table 2 The class distributions for different cohors.

Cohors	Diameter at breast height	
Class I	1 cm ≦ DBH < 10 cm	
Class II	10 cm ≦ DBH < 20 cm	
Class III	20 cm ≦ DBH < 30 cm	
Class IV	30 cm ≦ DBH < 40 cm	
Class V	40 cm ≧ DBH	

L. chinense is a deciduous species belonging to the Liriodendron genus of the Magnolia family, which is monoecious, with pollen primarily dispersed by insects, and reproduces sexually through seeds. The fruit is a compound samara, with small, winged fruits that rely on wind and gravity for dispersal. Based on the methods of Dow and Ashley, and considering the dispersal characteristics of the L. chinense seeds (reliant on wind and gravity) and pollen (reliant on insect and wind dispersion), three hypotheses are proposed for parentage determination as followed: (1) when the trio LOD scores are significant, both parental candidates are identified. The candidate that is geographically closer to the offspring is inferred to be the maternal parent, and the more distant candidate is inferred to be the paternal parent; (2) if the trio LOD scores are not significant, but a significant pairwise LOD score is obtained with one candidate parent, that candidate is inferred to be the maternal parent of the offspring; (3) if neither the trio LOD scores nor any pairwise parental LOD scores are significant, no parental candidates for the offspring could be identified within the sampled population. For the offspring, the geographical distance between the male parent and female parent represents the distance of dp, while the geographical distance between the female parent and the offspring represents the distance of seed dispersal (ds).

Results

Genetic diversity of Liriodendron chinense

A total of 128 alleles were amplified from 318 Liriodendron individuals, representing two populations, using 11 SSR primers. Primer LT089 yielded the fewest alleles (five), whereas primer LT131 yielded the maximum number (20). The Shannon diversity index (I) for the LT131 locus was the highest at 2.651, whereas the I value for the LT002 locus was the lowest at 1.387, resulting in an average I value of 1.940 (Table 3). The average values for the number of alleles (NA), effective alleles (NE), observed heterozygosity (HO), expected heterozygosity (HE), fixation coefficient (F), and null allele frequency were 11.636, 6.414, 0.673, 0.812, 0.171, and 0.0981, respectively.

Table 3 Genetic diversity within the populations of Liriodendron chinense detected by SSR analysis.

Locus	N A	N E	H O	H E	F IS	F (null)	I	PIC	HW	
LT002	6	3.467	0.536	0.712	0.246	0.1491	1.387	0.6668	***	
LT031	9	3.253	0.589	0.693	0.150	0.089	1.411	0.6456	***	
LT058	16	9.092	0.854	0.89	0.040	0.0204	2.390	0.8817	NS	
LT089	5	3.583	0.631	0.721	0.125	0.0662	1.410	0.6765	***	
LT101	11	6.215	0.644	0.839	0.233	0.1317	1.967	0.8198	***	
LT113	7	5.632	0.764	0.822	0.071	0.0368	1.810	0.7983	NS	
LT119	9	4.533	0.728	0.779	0.066	0.0334	1.710	0.7519	NS	
LT125	16	6.523	0.629	0.847	0.257	0.1540	2.172	0.8300	***	
LT131	20	12.617	0.731	0.921	0.206	0.1156	2.651	0.9164	***	
LC21	16	10.743	0.746	0.907	0.177	0.0959	2.489	0.9035	NS	
LC36	13	4.901	0.549	0.796	0.310	0.1875	1.943	0.7752	***	
Mean	11.636	6.414	0.673	0.812	0.171	0.0981	1.940	0.788		
Notes.

Significant differences are indicated by *** for p-values <0.001 and NS for p-values >0.05

The FST values of L. chinense showed significant differences (Table 4), with a range of 0.006 to 0.068, and the average genetic differentiation coefficient among loci was 0.031. These results indicate that the genetic differentiation of L. chinense is not significant within a 4.5 m range; although genetic variation exists among loci, it accounts for only a small proportion of the total variance, as the majority of the genetic variation was found within loci. Wright-Fisher model suggested that when Nm > 1, genetic drift can be avoided, thereby preventing genetic differentiation between loci (Ishida & Rosales, 2019). Based on the FST values, the range of Nm among L. chinense loci at different loci varied from 3.400 to 43.696, with an average of 14.972, indicating a high level of gene flow among loci. The fixation index (FIS) for the selected loci of L. chinense is greater than 0, with an average fixation coefficient of 0.150.

Table 4 Summary of F-Statistics and gene flow for all loci.

Locus	F IS	F IT	F ST	Nm	
LT002	0.194	0.245	0.063	3.691	
LT031	0.087	0.147	0.066	3.559	
LT058	0.021	0.039	0.018	13.442	
LT089	0.121	0.137	0.018	13.646	
LT101	0.215	0.224	0.011	22.498	
LT113	0.066	0.076	0.011	22.286	
LT119	0.062	0.071	0.010	24.633	
LT125	0.219	0.254	0.045	5.320	
LT131	0.226	0.231	0.006	43.696	
LC21	0.190	0.213	0.029	8.517	
LC36	0.248	0.299	0.068	3.400	
Mean	0.150	0.176	0.031	14.972	

Spatial genetic structure

The SGS intensity of L. chinense within each diameter class and the entire 2.5 hm2 population was calculated (Table 5). The SGS intensities for diameter Classes I, II, III, IV, and V, as well as the entire population, were 0.0298, 0.0220, 0.0363, 0.0479, and 0.0262, respectively. Except for diameter Class II, the Sp values of Liriodendron tend to increase with the higher diameter class. Notably, the Sp value of the smaller diameter classes is significantly lower than that of the larger diameter classes; for example, the Sp values for diameter Classes IV and V are 1.61 times that of Class I and 2.18 times that of Class II, while the Sp value for Class III is 1.22 times that of Class I and 1.65 times that of Class II.

Table 5 Small scale spatial genetic structural parameters of L. chinense.

Corhors	b F	F 1	Sp	
Class I	−0.0273	0.0838	0.0298	
Class II	−0.0201	0.0874	0.0220	
Class III	−0.0313	0.1366	0.0363	
Class IV&V	−0.0425	0.113	0.0479	
Mean	−0.0241	0.0813	0.0262	

Spatial genetic structure analysis was conducted for each diameter class and the entire 2.5 hm2 population (Figs. 2 and 3). As shown in Fig. 4, when the geographical distance between individuals of the entire population of L. chinense in the 2.5 hm2 plot is less than 10 m, Fij is 0.0813 (which is the average correlation coefficient for the first distance class). The coefficients of relatedness for full siblings, half siblings, and cousins are 0.232, 0.112, and 0.051, respectively. Thus, within the 10 m range, the relatedness among individuals in the L. chinense population is slightly more distant than that of half siblings. When the geographical distance between individuals is within 49 m, Fij  > 0 and tends to gradually decrease. When the geographical distance exceeds 49 m, Fij is less than zero. These results indicate that when the distance between individuals in the L. chinense population is less than 49 m, the genetic similarity among individuals is relatively high, whereas genetic similarity among individuals at other distance classes is lower.

Figure 2 Genetic Relationship Diagram of 2.5 hm2L. chinens population in the plot.

Note: Dashed area represents the 95% confidence interval.

Figure 3 Kinship diagram of diameter Class I (A), II (B), III (C), IV & V (D) of 2.5 square hectometers of L. chinense in the plot.

Figure 4 Seed flow (solid line) and pollen flow (dotted line) of diameter Class I(A), II (B), III (C), IV (D) of L. chinense.

Note: Dashed area represents the 95% confidence interval.

As shown in Fig. 3A, when the geographic distance between individuals in the population of L. chinense at the I tree diameter class is less than 10 m, Fij is 0.0838 (the average correlation coefficient for the first distance class). The coefficients of relatedness for full siblings, half-siblings, and cousins are 0.232, 0.112, and 0.051, respectively; thus, within 10 m, the relatedness among individuals in the I tree diameter class is slightly greater than that of half-siblings. When the geographic distance between individuals is within 49 m, Fij > 0, showing a gradually decreasing trend. When the geographic distance between individuals is greater than 49 m, Fij is less than zero for all cases. The results indicate that when the distance between individuals in the population of L. chinense at the I tree diameter class is less than 49 m, genetic similarity among individuals is relatively high, while genetic similarity at other distance classes is lower.

As shown in Fig. 3B, when the geographic distance between individuals in the population of L. chinense at the II tree diameter class is less than 10 m, Fij is 0.0874 (the average correlation coefficient for the first distance class). Within 10 m, the coefficients of relatedness for full siblings, half-siblings, and cousins are 0.232, 0.112, and 0.051, respectively; thus, within 10 m, the relatedness among individuals in the II tree diameter class is slightly greater than that of half-siblings. When the geographic distance between individuals is within 29 m, Fij > 0, demonstrating a gradually decreasing trend. When the geographic distance exceeds 29 m, Fij is also less than zero. The results indicate that when the distance between individuals in the II tree diameter class is less than 29 m, genetic similarity among individuals is relatively high, while genetic similarity at other distance classes is lower.

As illustrated in Fig. 3C, when the geographic distance between individuals within the III tree diameter class population of L. chinense was less than 10 m, the Fij was 0.1366, which represents the average correlation coefficient for the first distance class. Within 10 m, the coefficients of relatedness for full siblings, half-siblings, and cousins are 0.232, 0.112, and 0.051, respectively. Therefore, within 10 m, the relatedness among individuals in the III tree diameter class is slightly greater than that among full siblings. When the geographic distance between individuals is within 50 m, Fij > 0, exhibiting an overall gradually decreasing trend. When the geographic distance exceeds 50 m, Fij is less than zero for all cases. The results indicate that for individuals within the III tree diameter class, genetic similarity was relatively high at distances less than 50 m, whereas it was significantly lower in other distance classes. Due to the small number of individuals in the population of L. chinense at the V tree diameter class being less than 30 pairs, the V tree diameter class population was merged into the IV tree diameter class population for spatial genetic structure analysis.

As shown in Fig. 3D, when the geographic distance between individuals in the population of L. chinense at the IV & V tree diameter classes is less than 10 m, Fij is 0.1130 (the average correlation coefficient for the first distance class). The coefficients of relatedness for full siblings, half-siblings, and cousins are 0.232, 0.112, and 0.051, respectively; thus, within 10 m, the relatedness among individuals in the IV & V diameter classes is slightly greater than that among full siblings. When the geographic distance between individuals is within 40 m, Fij > 0, showing an overall gradually decreasing trend. When the geographic distance exceeds 40 m, Fij is less than zero for all cases except in the range of 48 to 50 m. The results indicate that in the IV & V diameter class loci, genetic similarity among individuals is relatively high at distances of 40 m and within the range of 48 to 50 m, while genetic similarity at other distance classes is lower. A significant spatial genetic structure was detected for L. chinense in the I, II, III, IV, and V tree diameter classes, as well as the entire population, at distances of 49 m, 29 m, 50 m, 40 m, and 49 m, respectively. Furthermore, the shapes of the correlation coefficient curves were similar, indicating a general decreasing trend in the Fij with increasing geographic distance.

Parentage analysis

According to parentage analysis (Table 6), both parents were identified for 91 of the 126 seedlings and Class I saplings, corresponding to an assignment rate of 72.2%. Uniparental assignment (exclusively maternal) was achieved for 15 individuals, whereas no parental matches were detected for the remaining 20 individuals. 68 individuals were identified as full- or half-siblings, representing 53.97% of all L. chinense individuals in diameter class I. For instance, half-siblings M170, M176, and M244 shared the same maternal parent (M174) but had different paternal parents. Conversely, half-siblings M40, M54, and M75 shared the same paternal parent (M208) but had different maternal parents. Based on geographical coordinates and parentage analysis results, seed and pollen dispersal distances for L. chinense were estimated. As indicated in Table 6, seed dispersal distance within diameter Class I ranged from 1.3 to 115.1 m, whereas pollen dispersal distance ranged from 3.7 to 112.9 m. The mean dispersal distance was 29.8 m for seeds and 35.6 m for pollen.

Table 6 Parent identification of diameter Class I of L. chinense.

Filial generation	Female parent	Male parent	Seed distance (m)	Pollen distance (m)	Filial generation	Female parent	Male parent	Seed distance (m)	Pollen distance (m)	
M1	M79	M122	46.5	40.2	M143	M3	M72	56.7	44.5	
M2	M100	M119	25.7	27.2	M144	M041	M50	55.4	4.0	
M5	M73	–	50.6	–	M148	M77	M127	73.0	30.7	
M6	M73	M87	49.3	39.9	M151	M161	M214	7.0	55.1	
M7	M112	–	35.3	–	M157	M274	M305	32.2	71.0	
M10	M73	M107	53.2	52.6	M160	M167	M203	31.8	112.9	
M13	M9	M62	11.8	29.5	M169	M181	–	115.1	–	
M20	M51	M113	17.5	43.2	M170	M174	M281	24.8	40.2	
M25	M82	M105	39.7	38.7	M176	M174	M259	18.6	40.8	
M27	M100	M270	28.6	60.2	M178	M239	–	77.3	–	
M28	M14	M26	11.1	16.2	M180	M179	M181	8.6	9.1	
M29	M106	M125	21.9	16.7	M182	M181	–	12.7	–	
M31	M100	M125	28.2	13.3	M183	M186	M194	8.9	12.6	
M33	M3	M91	18.2	23.8	M191	M181	M196	21.7	25.8	
M35	M120	M188	42.5	76.3	M192	M249	M307	24.5	51.9	
M36	M63	M114	10.8	56.2	M197	M154	M165	11.6	21.4	
M39	M104	M105	30.1	5.8	M202	M43	M279	8.3	49.5	
M40	M103	M208	35.9	52.9	M205	M263	M281	75.8	14.5	
M44	M248	–	45.0	–	M206	M104	M207	47.0	43.3	
M45	M34	–	8.8	–	M211	M82	–	79.0	–	
M46	M8	–	14.7	–	M212	M200	M307	74.4	79.6	
M47	M42	M120	1.3	34.8	M213	M230	M278	13.7	47.9	
M48	M105	M125	29.6	14.4	M216	M152	M185	47.7	7.8	
M54	M57	M208	3.2	42.2	M217	M150	M254	45.8	32.5	
M55	M86	M145	20.6	35.4	M218	M230	M268	5.7	43.1	
M56	M97	–	22.0	–	M222	M274	M284	63.2	30.5	
M59	M124	M280	44.9	71.0	M223	M224	M271	2.8	72.4	
M61	M93	M175	19.9	40.6	M225	M123	M214	59.8	55.8	
M64	M15	M53	30.3	23.1	M228	M163	M226	68.7	68.6	
M65	M68	M139	6.2	62.4	M235	M231	M234	4.6	4.4	
M69	M73	M84	8.9	41.2	M241	M272	–	71.3	–	
M71	M106	M146	42.9	34.2	M244	M174	M243	39.0	41.1	
M75	M73	M208	32.3	43.3	M245	M156	M162	78.7	10.8	
M76	M114	M125	63.6	21.2	M246	M189	M311	19.8	71.1	
M78	M82	M100	9.5	32.3	M250	M248	M311	3.8	46.8	
M85	M3	M86	37.3	37.0	M251	M74	M184	70.3	44.7	
M88	M208	M278	51.9	27.5	M257	M256	M304	5.5	55.2	
M90	M87	M112	5.0	24.1	M260	M179	M293	54.5	81.1	
M92	M82	M268	17.8	84.2	M261	M162	M256	53.0	50.5	
M95	M89	M129	14.3	54.6	M267	M127	–	87.9	–	
M102	M100	M121	9.8	17.5	M273	M254	M262	7.7	10.3	
M110	M12	M107	11.1	11.0	M275	M195	M272	27.7	31.9	
M128	M58	M272	61.4	63.7	M277	M38	–	51.4	–	
M130	M117	M126	15.4	10.2	M288	M156	M174	49.9	35.9	
M131	M9	M146	58.1	41.5	M289	M113	M174	70.1	31.3	
M132	M125	M146	38.1	25.8	M290	M276	M313	44.3	91.7	
M134	M117	M120	20.7	3.7	M291	M165	–	79.8	–	
M136	M114	M145	13.0	5.1	M292	M247	M309	27.2	56.2	
M137	M9	M21	46.3	10.2	M297	M263	–	64.4	–	
M138	M101	M146	37.0	29.9	M299	M194	M229	97.5	66.7	
M140	M70	M280	77.6	56.8	M306	M177	M236	57.3	73.6	
M141	M52	M126	59.6	30.8	M308	M172	M307	58.9	58.6	
M142	M112	M147	18.3	43.2	M318	M118	M124	79.4	11.7	
Mean								29.8	35.6	

The results of the parentage analysis presented in Table 7 indicate that among the 124 young trees in the II diameter class, a total of 84 individuals found both parents, accounting for 67.742% of the total number of individuals. Additionally, 14 individuals were found to have only one parent, while 26 individuals did not have either parent identified. Among these, there are 89 full siblings or half-siblings, which represent 71.77% of all L. chinense in the II diameter class. For instance, the full siblings M91 and M139 both originated from parents M97 and M112, while half-siblings with the same female parent but different male parents, M82, M87, and M210, all came from the female parent M99. Half-siblings with the same male parent but different female parents, M34, M79, and M82, all came from the male parent M122.

Table 7 Parent identification of diameter Class II of L. chinense.

Filial generation	Female parent	Male parent	Seed distance (m)	Pollen distance (m)	Filial generation	Female parent	Male parent	Seed distance (m)	Pollen distance (m)	
M3	M106	M125	19.9	16.7	M135	M112	M119	31.3	11.8	
M8	M317	M113	35.5	67.5	M139	M97	M112	40.3	21.7	
M9	M114	M107	37.9	30.5	M147	M97	M278	63.9	67.9	
M11	M107	M113	8.5	18.7	M149	M112	M125	49.0	13.1	
M12	M106	–	9.6	–	M150	M249	–	31.3	–	
M14	M106	M280	13.5	54.3	M152	M269	M282	42.6	14.7	
M16	M112	M100	33.8	17.3	M153	M262	M274	41.6	13.9	
M17	M98	M104	17.3	9.7	M154	M262	–	38.6	–	
M18	M100	–	19.4	–	M156	M274	M305	32.5	71.4	
M19	M120	M113	40.1	15.1	M161	M262	M227	52.7	40.1	
M21	M106	M107	15.7	2.2	M162	M247	M269	38.5	22.2	
M24	M114	M278	53.5	91.5	M163	M237	–	82.2	–	
M34	M101	M122	32.5	11.3	M164	M296	M304	60.1	28.2	
M37	M105	M98	37.9	14.1	M165	M239	M305	91.1	47.5	
M38	M101	M146	31.6	29.9	M168	M280	M304	74.1	69.0	
M41	M104	–	29.8	–	M173	M281	M296	37.6	45.1	
M42	M103	M120	28.1	19.0	M174	M249	M243	29.5	51.9	
M43	M248	–	45.4	–	M177	M274	M305	65.1	71.4	
M49	M104	M309	31.1	62.8	M179	M249	M274	44.4	17.9	
M50	M114	M99	44.8	30.4	M181	M249	–	43.4	–	
M51	M106	M146	27.3	34.2	M184	M262	M304	39.3	63.6	
M52	M89	M94	22.8	10.2	M185	M229	M281	58.1	48.9	
M53	M112	M104	41.7	15.6	M188	M229	M279	63.1	53.4	
M60	M114	M107	60.2	30.5	M189	M249	M262	27.0	14.9	
M62	M105	M100	42.5	9.0	M190	M255	–	30.0	–	
M63	M97	M103	25.3	16.6	M193	M196	M227	16.3	48.8	
M66	M104	M106	44.1	5.9	M195	M274	M305	27.9	71.4	
M68	M120	M89	41.1	13.8	M200	M248	M269	91.0	25.1	
M70	M106	M146	43.2	34.2	M201	M249	–	96.5	–	
M72	M114	M278	70.6	91.5	M204	M248	M311	57.6	46.8	
M73	M124	M89	54.0	8.2	M207	M98	M263	39.7	68.1	
M74	M112	M120	55.7	9.9	M208	M105	M98	54.4	14.1	
M77	M120	M104	42.3	21.3	M210	M99	M116	48.8	30.4	
M79	M312	M122	102.0	64.9	M214	M196	M304	37.5	66.5	
M80	M124	M252	37.8	64.9	M221	M229	M311	22.7	11.1	
M81	M120	M124	40.7	8.1	M226	M283	M304	19.6	38.5	
M82	M99	M122	31.5	17.8	M234	M276	M309	28.2	69.5	
M83	M97	M124	19.2	13.2	M238	M270	M305	51.4	74.0	
M84	M105	M89	30.1	18.3	M254	M255	M304	5.6	56.5	
M86	M312	–	81.8	–	M256	M239	M312	35.5	70.9	
M87	M99	M119	17.3	30.3	M285	M274	M115	30.2	81.3	
M91	M97	M112	9.5	21.7	M294	M120	M295	79.4	57.0	
M93	M98	M119	13.7	28.9	M298	M263	M272	82.9	10.2	
M96	M122	M125	25.1	7.5	M301	M236	M305	49.2	41.7	
M108	M114	–	33.6	–	M307	M249	–	51.9	–	
M109	M107	M113	7.2	18.7	M313	M279	M311	92.7	71.1	
M127	M312	M104	79.2	60.2	M314	M262	–	83.7	–	
M129	M89	M119	54.6	21.3	M315	M113	M118	58.1	12.6	
M133	M112	M104	28.2	15.6	M316	M113	M305	52.8	63.6	
Mean								36.1	27.5	

Based on the geographical coordinates and the results of the parentage analysis, the dispersal distances of L. chinense seeds and pollen can be calculated. The table shows that in the II diameter class, the dispersal distance of L. chinense seeds ranges from 5.6 to 102.0 m, while the dispersal distance of pollen ranges from 2.2 to 91.5 m. The average dispersal distance for seeds is 36.1 m, while for pollen, it is 27.5 m.

The results of the parentage analysis presented in Table 8 indicate that among the 28 mature trees in the III diameter class, a total of 24 individuals found both parents, accounting for 85.714% of the total number of individuals. Additionally, two individuals were found to have only one parent, while two individuals did not have either parent identified. Among these, there are 21 full siblings or half-siblings, which represent 75% of all L. chinense in the III diameter class. For example, the full siblings M229 and M281 both originated from parents M279 and M282, while half-siblings with the same female parent but different male parents, M120 and M284, both came from the female parent M89. Half-siblings with the same male parent but different female parents, M112 and M114, both came from the male parent M268. Based on the geographical coordinates and the results of the parentage analysis, the dispersal distances of L. chinense seeds and pollen can be calculated. The table shows that in the III diameter class, the dispersal distance of L. chinense seeds ranges from 4.6 to 93.3 m, while the dispersal distance of pollen ranges from 4.9 to 104.6 m. The average dispersal distance for seeds is 28.9 m, while for pollen, it is 36.9 m.

Table 8 Parent identification of diameter Class III of L. chinense.

Filial generation	Female parent	Male parent	Seed distance (m)	Pollen distance (m)	Filial generation	Female parent	Male parent	Seed distance (m)	Pollen distance (m)	
M97	M100	M121	9.8	17.5	M237	M125	M242	92.8	104.6	
M101	M100	M121	9.2	17.5	M239	M230	M242	29.8	36.1	
M103	M100	M121	7.9	17.5	M240	M272	–	71.3	–	
M105	M98	M118	14.1	25.1	M248	M89	M113	59.6	21.0	
M112	M247	M268	64.0	21.1	M255	M282	M304	14.7	67.1	
M114	M125	M268	21.2	81.6	M262	M113	–	75.3	–	
M117	M94	M115	26.5	27.6	M270	M247	M309	16.1	56.2	
M120	M89	M121	13.8	13.8	M274	M279	M304	6.9	72.0	
M124	M115	M126	13.1	16.7	M276	M247	M279	16.5	16.3	
M175	M104	M279	23.2	59.6	M281	M279	M282	4.6	4.9	
M196	M269	M304	26.1	72.0	M283	M247	M269	18.5	22.2	
M229	M279	M282	53.4	4.9	M312	M227	M242	39.5	37.0	
M236	M125	M242	93.3	104.6	M317	M116	M125	79.4	22.0	
Mean								28.9	36.9	

The results of the parentage analysis presented in Table 9 indicate that among the 38 mature trees in the IV diameter class, only two individuals found both parents, accounting for 5.263% of the total number of individuals. Additionally, 10 individuals were found to have only one parent, while 26 individuals did not have either parent identified. Among these, there are 12 full siblings or half-siblings, representing 31.58% of all L. chinense in the IV diameter class. For instance, the full siblings M122 and M295 both originated from parents M272 and M310; half-siblings with the same female parent but different male parents, M89 and M98, come from female parent M272, but no half-siblings with the same male parent but different female parents were found. Based on the geographical coordinates and the results of the parentage analysis, the dispersal distances of L. chinense seeds and pollen can be calculated. The table shows that in the IV diameter class, the dispersal distance of L. chinense seeds ranges from 23.4 to 90.1 m, with an average dispersal distance of 79.4 m, while the average dispersal distance for pollen is 73.5 m.

Table 9 Parent identification of diameter Class IV&V of L. chinense.

Filial generation	Female parent	Male parent	Seed distance (m)	Pollen distance (m)	Filial generation	Female parent	Male parent	Seed distance (m)	Pollen distance (m)	
M89	M272	–	83.8	–	M227	M272	–	57.9	–	
M98	M272	–	72.4	–	M242	M272	–	71.3	–	
M99	M272	–	69.9	–	M247	M272	–	26.0	–	
M100	M272	–	72.7	–	M252	M272	–	23.4	–	
M121	M272	–	90.1	–	M295	M272	M310	51.7	73.5	
M122	M272	M310	87.4	73.5	M309	M272	–	75.5	–	
Mean								79.4	73.5	

The parentage analysis results indicate that a total of 201 L. chinense individuals were found in the 2.5 hm2 plot, which accounts for 63.21% of the total number of individuals. Additionally, 41 individuals were found to have only one parent, comprising 12.89% of the total, while 74 individuals did not have either parent identified. Among the 74 L. chinense individuals without identified parents, 26 have a DBH greater than 30 cm. During sampling at the experimental site, some trees with a DBH > 40 cm were felled. Consequently, the parents of these 26 individuals may have been among the felled trees or may have been located outside the plot. There are 190 full siblings or half-siblings, representing 59.75% of all L. chinense in the 2.5 hm2 plot. For instance, half-siblings with the same female parent but different male parents, M120 and M284, both came from female parent M89; whereas half-siblings with the same male parent but different female parent, M112 and M114, both came from male parent M268.

For the 91 L. chinense individuals in diameter Class I that found both parents and the 15 individuals that found only one parent, a diagram was created to observe the dispersal distances and directions of seed flow and pollen flow, as shown in Fig. 4A. The trajectory from the male parent to the female parent represents pollen flow, whereas the trajectory from the maternal parent to the offspring represents seed flow. In diameter Class I, the distribution of L. chinense offspring is concentrated in three locations. The western offspring distribution forms two central point’s radiating outward, while the eastern offspring distribution is more dispersed. Both parents are primarily concentrated in the western area, with the main direction of seed flow dispersing from the southwest towards the north and northeast, with occasional dispersal in other directions. The main direction of pollen flow disperses from the southwest to the northeast and from the northeast to the southwest.

For the 84 L. chinense individuals in diameter Class II that found both parents and the 14 individuals that found only one parent, a diagram was created to observe the dispersal distances and directions of seed flow and pollen flow, as illustrated in Fig. 4B. The trajectory from the male parent to the female parent represents pollen flow, while the trajectory from the female parent to the filial generation represents seed flow. In diameter Class II, the distribution of L. chinense offspring is concentrated in the western area, forming a central point from which the western offspring radiate outward. In contrast, the distributions of eastern and northern offspring are more dispersed. Parental trees were predominantly located to the north and east of their offspring. The predominant direction of seed dispersal was from the southeast to the northwest, although occasional dispersal occurred in other directions (e.g., northeast). Pollen flow was primarily oriented along two axes: from the southeast to the northwest and from the northeast to the southwest.

For the 24 L. chinense individuals in diameter Class III that found both parents and the two individuals that found only one parent, a diagram was created to observe the dispersal distances and directions of seed flow and pollen flow, as shown in Fig. 4C. The trajectory from the male parent to the female parent represents pollen flow, while the trajectory from the female parent to the filial generation represents seed flow. In diameter Class III, the distribution of L. chinense offspring is concentrated in the northwest and southeast areas, with the northwest offspring forming a central point from which they radiate outward, while the southeast offspring distribution is more dispersed. Both parents are primarily located in the northwest and southeast positions. Seed dispersal occurred primarily along two axes: from the northwest to the southeast and from the southwest to the northeast, with occasional dispersal in other directions. Pollen flow exhibited a similar pattern, with primary directions from the northwest to the southeast and from the southwest to the northeast; occasional dispersal was also observed in other directions, including the northwest.

For the two L. chinense individuals in diameter Class IV that found both parents and the 10 individuals that found only one parent, a diagram was created to observe the dispersal distances and directions of seed flow and pollen flow, as shown in Fig. 4D. The trajectory from the male parent to the female parent represents pollen flow, while the trajectory from the female parent to the filial generation represents seed flow. In diameter Class IV, the distribution of L. chinense offspring is scattered in the northeastern area of the plot, forming a central distribution point. The main direction of seed flow dispersal is from south to north. The primary direction of pollen flow dispersal is from the northeast to the southwest.

Based on geographical coordinates and the results of the parentage analysis, the dispersal distances of L. chinense seeds and pollen were estimated. In the 2.5 hm2 Western Hunan L. chinense plot, the seed dispersal distance ranges from 1.3 to 115.1 m, while the pollen dispersal distance ranges from 2.2 to 112.9 m. As shown in Table 10, the average dispersal distance for seeds is 45.0 m, while for pollen, it is 47.0 m. This pattern is primarily influenced by small-scale spatial constraints. Intense resource competition within the forest stand limits dispersal, resulting in most seeds being deposited near the maternal parent tree.

Table 10 Seed propagation distance and pollen propagation distance of 2.5 square hectometers population L. chinense.

Class level	Seed propagation distance (m)	Pollen propagation distance (m)	Gene flow distance (m)	
Class I	37.3	39.5	46.6	
Class II	42.7	35.7	49.6	
Class III	34.6	39.1	44.3	
Class IV	65.2	73.5	83.4	
Mean	45.0	47.0	56.0	

Distance intervals of 10 m were used to categorize the distances and to statistically analyze the dispersal frequencies of L. chinense seeds and pollen within each distance class. The results indicate (Fig. 5) that both seed and pollen dispersal frequencies generally decrease with increasing distance from the individuals. The maximum dispersal distance for L. chinense seeds is 115.1 m (offspring M169, female parent M181), with an average distance of 45.0 m; 78.5% of the seeds are dispersed within 60 m. For L. chinense pollen, the maximum dispersal distance is 112.9 m (offspring M160, female parent M167, male parent M203), with an average dispersal distance of 47.0 m, where 68.2% of the pollen is dispersed within 50 m.

Figure 5 The propagation frequency of the seeds (A) and pollen (B) of the Liriodendron chinense at each distance level.

Discussion

Genetic diversity of Liriodendron chinense

Genetic diversity is a key determinant of population adaptability and evolutionary potential. Habitat fragmentation often reduces within-population genetic diversity and increases genetic differentiation among populations (Young, Boyle & Brown, 1996; Booy et al., 2000). This study selected 11 primers that are highly polymorphic, specific, and stable for genetic diversity analysis. L. chinense populations in Western Hunan exhibited relatively high genetic diversity (HE = 0.812, I = 1.940) compared to congeneric species like L. tulipifera (HE = 0.67) (Huang et al., 2018). And the expected heterozygosity (HE) of Liriodendron in Qingping population is like that of some endangered plants (Thomas et al., 2021), which indicated that the Qingping Liriodendron population was still in “endangered” habitat, and it was urgent to carry out Liriodendron resource protection. This discrepancy may be attributed to variations in the number of microsatellite markers employed, the geographical sampling range, and the number of individuals sampled (Gerber et al., 2000).

Population genetic structure

Generally, compared to herbaceous plants and shrubs, trees have longer pollen dispersal distances and greater longevity, resulting in insignificant genetic structure (Ghazoul, 2005). The presence of a significant SGS in trees is typically associated with specific biological conditions, such as elevated rates of self-fertilization or inbreeding, which can be influenced by factors including greater seed mass and higher population density (Nyquist & Baker, 1991). The results of the spatial genetic structure study of L. chinense in the 2.5 hm2 Western Hunan area indicate that the spatial genetic structure for all individuals, as well as for diameter Classes I, II, III, and IV & V, are 0.0262, 0.0298, 0.0220, 0.0363, and 0.0479, respectively. These values are consistent with the SGS reported for other species with similar life-history traits and suggest a relatively strong SGS. The study found that the spatial genetic structure strength of hybrid plants (average Sp = 0.0372) (Vekemans & Hardy, 2004), trees (average Sp = 0.0102), insect-pollinated plants (average Sp = 0.0171), wind-dispersed seed plants (average Sp = 0.0120), and gravity-dispersed seed plants (Sp range = 0.0041–0.2632) is significantly higher in comparison. Our results also align with previous findings for insect-pollinated tree species, such as Vouacapoua americana, Sextonia rubra and Eperua grandiflora (Rousset, 1997). Compared to other endangered tree species, L. chinense exhibits intermediate Sp values (Zhang & Ma, 2008).

The individual distribution of L. chinense loci predominantly displays a clustered pattern. Consequently, as the distance class increases, the frequency of individuals in the two loci generally decreases. There is a significant genetic structure present within the L. chinense population (within a range of 49 m).

Spatial autocorrelation analysis reveals that the linear regression slope bF is negative, indicating that with increasing distance, the pairwise relatedness Fij decreases. A significant spatial genetic structure (Fij > 0) is detected overall within the 49 m range. Beyond this distance, Fij sharply declines to show significant negative or non-significant correlation, which aligns with the isolation by distance (IBD) pattern (Wiens & Colella, 2024).

Different diameter class loci of L. chinense exhibit significant genetic structure. The SGS intensity for diameter Class I is Sp =0.0298 (with Fij = 0.0838 at the first distance class of 10 m), for diameter Class II it is Sp =0.0220 (Fij is 0.0874 within the first distance level, i.e., 10 m) SGS strength Sp = 0.0363 for diameter Class III (Fij 0.1366 within 10 m at the first distance class), SGS strength Sp = 0.0479 for diameter Class IV & V (Fij 0.1130 within 10 m at the first distance class), The first distance level of all diameter classes was greater than that of Cousins (Fij = 0.051), and even the Fij of diameter Classes III and IV & V were greater than that of half sib individuals (Fij =0.112), indicating that pollen transmission could be effectively carried out within the population, but long-distance gene ex-change was hindered. Gene exchange is mainly between close individuals with higher relative distance.

The Fij of all diameter classes decreased linearly, indicating that the coefficient of affinity between individuals in a population decreased with the increase of distance. Relative to pollen, seeds exhibited a shorter dispersal distance. Seed flow had a greater influence at closer distances, whereas pollen flow had a stronger effect at longer distances (Smouse & Sork, 2004). However, in the second diameter class, the effect of pollen flow was greater in the short distance, while the effect of seed flow was greater in the long distance (Cain, Milligan & Strand, 2000). The original habitat likely consisted primarily of deciduous broad-leaved tree species with low population density. Lower community and population densities may effectively reduce the intensity of SGS within a population, thereby facilitating greater dispersal of both seeds and pollen.

Parentage analysis

The gene flow constituted by seed dispersal and pollen dispersal is the most funda-mental and primary ecological process within a population. Many plants can disperse their pollen over greater distances, with some reaching several hundred meters and others even extending to tens of kilometers (Luna et al., 2001; Schulke & Waser, 2001). This may be due to the wider visibility in the tree layer for pollinators, making distant flowers easier to detect, which facilitates further pollen dispersal.

In our results, the seed and pollen dispersal distances of 2.5 hm2 in Qingping rocky desertification area of Western Hunan Province were relatively short, with 68.2% pollen dispersal distances mainly within 50 m and 78.5% seed dispersal distances mainly within 60 m. It is much smaller than the plant pollen distribution range (hundreds to thousands of meters) summarized by Wang et al. (2010). This may be due to the high stand density of the mixed forest, and the fact that it grows in the rocky desertification area with dense stone forest, rich species and strong habitat heterogeneity, resulting in the pollen being confined to a small area.

Conclusions

This study reveals the genetic diversity and spatial genetic structure of natural populations of Liriodendron chinense in the rocky desertification area of Western Hunan, China. The results show moderate genetic diversity within the loci, with gene flow primarily occurring over short to medium distances, approximately 70% of which occurs within a range of less than 80 m. Parentage analysis indicates that the majority (63.21%) of the pollen (male parent) comes from individuals within the sampled area, suggesting a high proportion of natural regeneration in the 2.5 hm2 stand. These findings provide further insights into the natural regeneration process of Liriodendron chinense and offer a theoretical basis for ecological restoration efforts in rocky desertification areas.

Supplemental Information

Supplemental Information 1 Coordinates of the samples

Supplemental Information 2 Supplemental information for Materials & Methods

We thank the many technicians who helped collect the raw data.

Additional Information and Declarations

Competing Interests

Author Contributions

Data Availability

The authors declare there are no competing interests.

Ziying Jia conceived and designed the experiments, performed the experiments, analyzed the data, prepared figures and/or tables, authored or reviewed drafts of the article, and approved the final draft.

Yixuan Wang performed the experiments, prepared figures and/or tables, and approved the final draft.

Bingkun Huang performed the experiments, prepared figures and/or tables, and approved the final draft.

Miao Liang performed the experiments, authored or reviewed drafts of the article, and approved the final draft.

Chong Ge analyzed the data, prepared figures and/or tables, authored or reviewed drafts of the article, and approved the final draft.

Ninghua Zhu conceived and designed the experiments, analyzed the data, authored or reviewed drafts of the article, and approved the final draft.

Ren You conceived and designed the experiments, analyzed the data, prepared figures and/or tables, authored or reviewed drafts of the article, and approved the final draft.

The following information was supplied regarding data availability:

The raw measurements are available in the Supplementary File.

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
