# Peer review of "Genetic diversity analysis of the natural regeneration loci of Liriodendron chinense in artificial mixed forests in the rocky desertification area of Western Hunan"

_PeerJ, doi:10.7717/peerj.20138_

## Round 0.1 · original submission · Major Revisions

Dear Dr. You, I ask you to correct the shortcomings pointed out by the reviewers. I am optimistic that the new version of this article will be approved by the reviewers and will be accepted for publication.

**Language Note:** The review process has identified that the English language must be improved. PeerJ can provide language editing services - please contact us at [email protected] for pricing (be sure to provide your manuscript number and title). Alternatively, you should make your own arrangements to improve the language quality and provide details in your response letter. – PeerJ Staff

·

Basic reporting

There are minor English usage revisions needed throughout the manuscript
- In Lines 30-31, authors might need to revise the sentence and ensure the intended meaning is retained (L. chinense with a large diameter can reproduce and can be considered potential parents).
- In Line 35, does the 41 resemble the female parent part of the 201? This sentence requires a clear description.
- The authors have used the term ‘L. chinensis’ in Line 84. I advise them to maintain L. chinense.
- Lines 82-86 should be rephrased may be too wordy and long, and a reader might lose the intended meaning.
- In Line 96, the authors might need to observe the use of a period. In addition, in this sentence to line 98; the sentence structure might be rephrased. Maybe to something like this;
We measured the breast height (DBH)… to record the../ To record the …, we measured…
- Still, the authors might need a thorough review of several sentence structures on this above note.
- Is Line 130 a subheading or complete sentence?

Experimental design

- The last three subsections of the Methodology section are lengthy and wordy. If these subsections can not be reduced, I suggest to the others to further make more subsections of each or restructure them.
- One of the major delimits in the methodology description is most sentences lacked connectivity; information dissemination was not flowing. Felt like a simple stating of each statement. The authors should revise and disseminate the information to flow.
- In Lines 116-124, is the PCR procedure needed in this part? To reduce content in this section, the authors can add this procedure in the Additional file.
- In Lines 146, 148, and 195 the authors make mention of the olive tree, is it still the L. chinense?
- In Line 164, the Class distributions can be better presented as a table.

Validity of the findings

- In my opinion, the Result section is well-elaborated, although some parts are too wordy. Maybe some descriptions like
- Line 229 lacks reference.

All the Best!

Additional comments

This manuscript investigates the evolutionary dynamics of the Liriodendron chinense, analyzing its genetic diversity and spatial genetic structures using SSR primers. In my opinion, this research is crucial and helps perpetuate the L. chinense tree species while benefiting from it ecologically.

Overall, while the manuscript presents interesting findings, addressing the methodological concerns outlined above is crucial before publication. I recommend 'Minor Revisions' to improve clarity and reproducibility.

·

Basic reporting

The authors did relatively good work for the Liriodendron chinense population to measure genetic diversity, and seed and pollen dispersal. I have read through the entire manuscript, and I conclude that this work deserves to be published in this journal. However, I have general comments and suggestions here. The authors must check English grammar and rewrite the whole MS. I see there are plenty of issues with grammar. I notice that they always say “populations”. But I think they only choose one population. Sometimes they delimit two subpopulations by a 4.5 m distance which is not scientifically logical. I noticed also, that they described genetic indices between loci as between populations. It’s not the correct way to do that. They must describe between loci and choose average values to use to compare with other papers. I recommend them to be cautious with scientific terms and rules. There are almost no questions or comments about the methodology they used. Indeed, these methods are suitable for the desired aim of the work. However, the way that they are describing the results is not scientifically correct. I also suggest to rewrite the Discussion part entirely with comparing to relevant papers. My general suggestion is that please while rewriting be more cautious, consistent, and scientifically logical.

My other comments:

Lines 69–73 are not very accurate to understand, please make them more cohesive and logical.

Line 84, L. chinensis is not italic.

Lines 85–86, what is the difference between spatial genetic structure and genetic structure? Maybe duplicated?

Line 96, “And” should be “and”.

Lines 112–113, repeated sentences.

Line 138, you said tulip tree, and in line 146 you said olive tree. You need to clarify them. L. chinense is an olive tree or a tulip tree?

Lines 218–224, I think the order should be like this: The primer LT089 detected the fewest alleles, amplifying 5 alleles, while primer LT131 detected the most alleles, amplifying 20 alleles. The Shannon diversity index (I) for the LT131 locus was the highest at 2.651, whereas the I value for the LT002 locus was the lowest at 1.387, resulting in an average I value of 1.940 (Table 2). The average values for the number of amplified alleles (NA), effective alleles (NE), observed heterozygosity (HO), expected heterozygosity (HE), fixation coefficient (F), and null allele frequency were 11.636, 6.414, 0.673, 0.812, 0.171, and 0.0981, respectively.

Lines 225–226, you say FST between populations. But in the methodology as far as I understand, you choose only one population. In Table 3 you gave F statistics for loci. So you cannot say FST between populations, right? Because the term “population” is different from what you are trying to explain. I read in the entire manuscript you say “populations”. I think you need to change it to another term. Because you have only one population. You may refer to “loci” not populations.

Lines 308–311, please make it clear by using correct English grammar. For me, it took 10 minutes to understand what you mean here.

Line 453, Liriodendum?

Line 434, Genetic diversity of Liriodendron chinense, this section is written very badly, it seems like a discussion but is not organized well and logically. You need to improve it scientifically. You said a and b subpopulations. From where did you get this? Before you didn’t mention them, right? And, how 4.5 m distance can delimit population into subpopulations? I don’t think it’s logical. Please rewrite this section in a well-structured way and compare your results with other relevant papers. While comparing genetic indices please use average genetic indices, because you did it for one population.

Other parts of the discussion also should be rewritten in a well-structured way.

Experimental design

-

Validity of the findings

-

---

## Round 0.2 · Major Revisions

Dear Dr. You, I ask you to carefully revise the manuscript and give it to a native speaker for review. The manuscript contains many inconsistent, incomprehensible places for readers.

Julin Maloof, the Section Editor, has commented and said:

This manuscript needs significant editing for grammar and language usage. Because of the problems the manuscript is difficult to read and it is difficult to understand the authors' meaning in many places. I am only going to mention a few instances, BUT THE WHOLE MANUSCRIPT NEEDS REVISION, not just what I am pointing out. I recommend a professional scientific editing service be used +++ There seems to be misuse of the word "loci". It is sometimes used correctly to indicate genetic locations in the genome. But also it seems to be incorrectly used to indicate geographic location: 57 "in its natural loci" (or maybe they meant "natural genetic diversity"?; 61 "Genetic diversity within loci" (possible okay if they are talking about allelic diversity); 88 "secondary loci" (what are secondary loci?). Plus what do the authors mean by "adult loci", "regeneration loci", "natural loci"? In genetic analyses "regeneration loci" might be taken to mean genes or loci that are causally associated with a plants ability to regenerate but that is not the case here. line 191 "two loci" what is this referring to? two locations? But only 1 locations is mentioned in the methods +++ I am not even sure what is meant by regeneration in this manuscript; do the authors just mean plants grown from seed? +++ line 35 "201 were identified as most as the parental line" poorly written, what does this mean? +++ line 28" (N = 318)". The way this is written makes it seem like 318 loci were studied, not 318 individuals. +++ line 31 "in the large diameter": diameter is an adjective; a noun is needed (e.g. "in the large diameter class"). +++ Many sentences are missing a subject e.g. line 29 "Conducted...", line 97 "Utilized", line 125 "Using" I think this is missing both a subject and verb. line 137 "Calculate", etc.
+++ 96 "real-time kinematic". Kinematic is an adjective it needs a noun after it.
+++ line 179 how can a tree offspring find its parents?
+++ line 235: I'm not an expert on Fij but I would have thought that Fij for full-sibs would be 0.25, not 0.232?
+++ line 80 "this study using" there is no verb.
+++ The abstract mentions that the plant "plays a crucial role in enhancing biodiversity" but not evidence or literature citation is provided to support this claim.
+++ What are the dashed lines in Figure 2?
+++ Figure 4 legend says it is of seed flow and pollen flow. Why are there two dashed lines? Why is the solid line labeled as Kij?
+++ Table 2 type "cohors"
+++ The results would benefit from a little more explanation about hypotheses/questions being tested in each paragraph.

**Language Note:** The Academic Editor has identified that the English language must be improved. PeerJ can provide language editing services - please contact us at [email protected] for pricing (be sure to provide your manuscript number and title). Alternatively, you should make your own arrangements to improve the language quality and provide details in your response letter. – PeerJ Staff

·

Basic reporting

The article was well revised, English usage was clear and it conforms to professional standards.

Experimental design

The authors made relevant revisions, now the article's experimantal design is well defined, relevant and meaningful.

Validity of the findings

This research is novelty, it provides new knowledge of the L. chinense. Additionally, this research has provided factual conclusions and sound results.

Additional comments

This manuscript presents a well-structured and clearly written study on Genetic Diversity of the Natural Regeneration Loci of Liriodendron chinense , which addresses an important question in the field of plant genetic diversity and evolution.

·

Basic reporting

Now it seems quite good. All the best!

Experimental design

good

Validity of the findings

good

Additional comments

good

---

## Round 0.3 · accepted · Accept

Dear Dr. You, I congratulate you on the acceptance of this article for publication.